# Quantitative Inversion of Lunar Surface Chemistry Based on Hyperspectral Feature Bands and Extremely Randomized Trees Algorithm

Shuangshuang Wu [1,2], Jianping Chen [1,2,*], Li Li [3], Cheng Zhang [1,2], Rujin Huang [1,2] and Quanping Zhang [1,2]

1   School of Earth Sciences and Resources, China University of Geosciences, Beijing 100083, China
2   Beijing Key Laboratory of Development and Research for Land Resources Information, Beijing 100083, China
3   College of Geomatics, Xi'an University of Science and Technology, Xi'an 710054, China
*   Correspondence: 3s@cugb.edu.cn

**Abstract:** In situ resource utilization (ISRU) is required for the operation of both medium and long-term exploration missions to provide metallic materials for the construction of lunar base infrastructure and $H_2O$ and $O_2$ for life support. The study of the distribution of the lunar surface elements (Fe, Ti, Al, and Si) is the basis for the in situ utilization of mineral resources. With the arrival of the era of big data, the application of big data concepts and technical methods to lunar surface chemistry inversion has become an inevitable trend. This paper is guided by big data theory, and the Apollo 17 region and the area near the Copernicus crater are selected for analysis. The dimensionality of the first-order differential spectral features of lunar soil samples is reduced based on Pearson correlation analysis and the successive projections algorithm (SPA), and the extremely randomized trees (Extra-Trees) algorithm is applied to Chang'E-1 Interference Imaging Spectrometer (IIM) data to establish a prediction model for the lunar surface chemistry and generate FeO, $TiO_2$, $Al_2O_3$, and $SiO_2$ distribution maps. The results show that the optimum number of variables for FeO, $TiO_2$, $Al_2O_3$, and $SiO_2$ is 17, 5, 8, and 30, respectively. The accuracy of the Extra-Trees model using the best variables was improved over that of the original band model, with determination coefficients ($R^2$) of 0.962, 0.944, 0.964, and 0.860 for FeO, $TiO_2$, $Al_2O_3$, and $SiO_2$, and root mean square errors (RMSEs) of 1.028, 0.672, 0.942, and 0.897, respectively. The modeling feature variables and model preference methods in this study can improve the inversion accuracy of chemical abundance to some extent, demonstrating the potential of IIM data in predicting chemical abundance and providing a good data basis for lunar geological evolution studies and ISRU.

**Keywords:** moon; machine learning; successive projections algorithm; extremely randomized trees; lunar chemistry

## 1. Introduction

From ancient times to the present day, humankind has never stopped exploring the vastness of the universe. The Moon, the closest celestial body to the Earth, has always been a prime target for human astronomical activities and is naturally the first step for humankind to step out of the cradle of the Earth and into the vastness of the universe [1]. As humankind is currently facing a shortage of resources, the exploitation and utilization of the rich lunar reserves of mineral resources can effectively alleviate the problem of resource shortages for humankind in the future. Given the high cost of Earth–Moon transportation, in situ resource utilization (ISRU) has become a fundamental technological guarantee for the establishment of a lunar base [2–4]. Currently, the most important ISRU project is to obtain $H_2O$ and $O_2$ for life support through ilmenite. However, increasing human understanding and knowledge of the moon has revealed that any oxide can also be reduced to its constituent elements through reduction processes and produce $H_2O$ and $O_2$ as byproducts, which is a key issue worthy of study [5]. Therefore, the elements of greatest

interest to us are the lunar surface elements Fe, Ti, Al, and Si. The study of the distribution of Fe, Ti, Al, and Si is the basis for assessing the prospects for the exploitation of lunar mineral resources and the rational exploitation of resources. Fe, Ti, Al, and Si can all be used as sources of building materials for lunar base construction, with Fe mainly existing in lunar mafic minerals, Ti accounting for a large proportion of the high-titanium mare basalt [6,7], almost all Al on the lunar surface found in highland plagioclase [8], and Si is always present in the form of silicates in nearly every rock and mineral grain on the Moon. Therefore, this paper focuses on the inversion of Fe, Ti, Al, and Si on the lunar surface to study the distribution and abundance of mineral resources and to lay the foundation for a better assessment of the lunar mineral resources.

Currently, multisource remote sensing techniques based on $\gamma$-ray, X-ray, and multi/hyperspectral data are widely used for lunar surface chemistry inversion [9–16]. Compared with high-energy techniques, multi/hyperspectral data have the advantage of high resolution; therefore, many studies invert chemistry abundances based on multi/hyperspectral data. The outcomes of these studies include FeO and $TiO_2$ distribution maps with a resolution of 100 m obtained using Clementine data [17]; FeO, $TiO_2$, $Al_2O_3$, $SiO_2$, CaO, and MgO distribution maps with a resolution of 200 m obtained based on IIM data [18–21]; FeO distribution maps with a resolution of 20 m obtained based on Moon Mineralogy Mapper ($M^3$) data [10]; a $TiO_2$ distribution map with a resolution of 400 m obtained from the Lunar Reconnaissance Orbiter Camera (LROC) Wide Angle Camera (WAC) [16]; and FeO, $TiO_2$, $Al_2O_3$, CaO, and MgO distribution maps with a resolution of 59 m obtained based on Kaguya Multiband Imager (MI) data [22,23].

Among the four elements found on the lunar surface, namely, Fe, Ti, Al, and Si, Fe and Ti are transition metals that can undergo ligand field transitions (d-d transitions). Accompanying this electronic transition from low to high energy levels, the transition metals $Fe^{2+}$ and $Ti^{3+}$ produce absorption features in the visible to near-infrared range [18,24–26]; thus, the spectral features of the lunar surface can be directly interpreted based on mineralogical principles [7,24,26]. Early quantitative spectroscopic analysis of lunar samples demonstrates the potential for inversion of lunar surface chemistry and mineral components using band ratio methods [27]. Currently, this approach has been widely used for FeO and $TiO_2$ inversion [16,17,19,28–32]. However, elemental abundance inversion is not limited to the chromophore elements Fe and Ti. Although nonchromophore elements do not have absorption characteristics in the visible NIR, they affect reflectance [18]. Therefore, many studies have used regression models to establish statistical relationships between elemental abundance and spectral characteristics to predict chemical abundance [20,21,33,34]. With continuous research, a variety of machine learning models have been applied for oxide inversion. Regardless of whether the input variables are full or feature bands, machine learning methods have displayed good performance in inversion [7,22,35–37]. For example, Korokhin et al. [7] proposed a nonlinear method based on an artificial neural network (ANN) inversion of $TiO_2$ and avoided problems related to the limited number of bands and the subjective selection of band combinations compared with the traditional linear regression method. Zhang et al. [37] used principal component analysis (PCA) combined with a support vector machine (SVM) to estimate the abundance of chemical compositions ($SiO_2$, $Al_2O_3$, FeO, MgO, and $TiO_2$) and their maturity indicators (Is/FeO), where PCA was used to downscale the reflectance spectra of lunar soil samples for screening and the SVM was used to build a predictive oxide content model. Wang et al. [22] used a particle swarm optimization-support vector machine (PSO-SVM) algorithm based on KAGUYA MI data to generate oxide abundance maps with a spatial resolution of up to 59 m/pixel, and the map was relatively free of topographic shadows.

The selection of appropriate modeling feature variables and machine learning models is particularly important for the accuracy of oxide inversion. Feature band selection methods can improve the efficiency and accuracy of machine learning models to some extent. However, methods that combine machine learning and multi/hyperspectral data for the inversion of the lunar surface oxide content do not fully consider the preferential selection of model feature variables. For example, Shkuratov [33] used multiple linear regression models for three bands of 750, 915, and 965 nm from SMART-1 data. Wang &

Niu [35] used 19 (571 nm–865 nm) bands of IIM data to construct a model after removing anomalous bands. Wu [18] selected 22 bands in the spectral range of 561–918 nm for modeling based on the magnitude of the signal-to-noise ratio. Sun et al. [20] screened four single bands and three band ratios with high correlations in the 513 nm–891 nm range based on Pearson correlation coefficients for modeling. Most of the above methods do not consider the interrelationships between pairs of spectral features or between oxides and spectral features, and adding all available bands to a model can have a negative impact on the accuracy and generalization ability of the model. In response to the above research deficiencies, Pearson correlation coefficients are adopted in this paper for the initial screening of spectral features. Then, input parameters are secondarily screened based on clustering analysis combined with the successive projections algorithm (SPA) to reduce the covariance interference among spectral features while ensuring a high correlation between input spectral features and oxides.

The extremely randomized trees (Extra-Trees) machine learning algorithm is selected to build the oxide inversion model and improve the oxide inversion accuracy. The Extra-Trees algorithm can effectively describe the small-sample, high-dimensional, and complex nonlinear relationship between the oxide content and spectral reflectance by integrating several weak learners to obtain a strong learner [38]. This approach shows a clear advantage in the inversion of the oxide content. For model sample set partitioning, Jin et al. [39], Li [27,40], and Zhang et al. [37] divided the training and test sets according to particle size. Zhou et al. [41] randomly selected 2/3 of the samples as the training set for modeling and 1/3 of the samples as the validation set. These segmentation methods are often random. In this paper, we use sample set partitioning based on the joint x-y distance (SPXY) algorithm to partition the samples in a way that maximizes the characterization of the sample distribution and improves the stability of the model.

The inversion of lunar surface chemistry based on the concept and analysis methods of big data is an inevitable trend in future development. The selection of suitable modeling feature variables and machine learning models is crucial for achieving high-accuracy oxide inversion. Guided by big data theory, big data concepts and technical methods are applied to lunar chemistry inversion, and a prediction method for feature band selection combined with an Extra-Trees model is proposed. First, the first-order difference bands are initially screened according to the Pearson correlation coefficient, and they are then downscaled using the bisecting K-means (BKM) algorithm combined with the SPA to determine the best band combination. Second, the Lunar Soil Characterization Consortium (LSCC) training sample is used to build an Extra-Trees prediction model. Finally, the model is applied to predict the oxide abundance in the Apollo 17 region of the lunar surface and the region near the Copernicus crater. The experimental results are compared with those of representative models [15,17,30,31], and it is found that the model shows good agreement with them, providing a new idea and method for the inversion of oxide abundances.

## 2. Materials and Methods

This paper proposes a method to invert the distribution of lunar surface chemistry, targeting the need for ISRU. Figure 1 shows the technological flow chart of this paper, which includes the following main steps:

(1) Feature band selection: The sensitive regions of each chemistry were initially screened according to Pearson correlation coefficients, and then clustering analysis combined with SPA was used for secondary screening to determine the best combination of bands.

(2) Construction of an Extra-Trees model: Seventy-six LSCC samples with reflectance and oxide content data were used as model inputs for training and testing.

(3) Prediction of chemical abundance: The IIM reflectance data were put into the model to estimate the lunar surface chemical abundance.

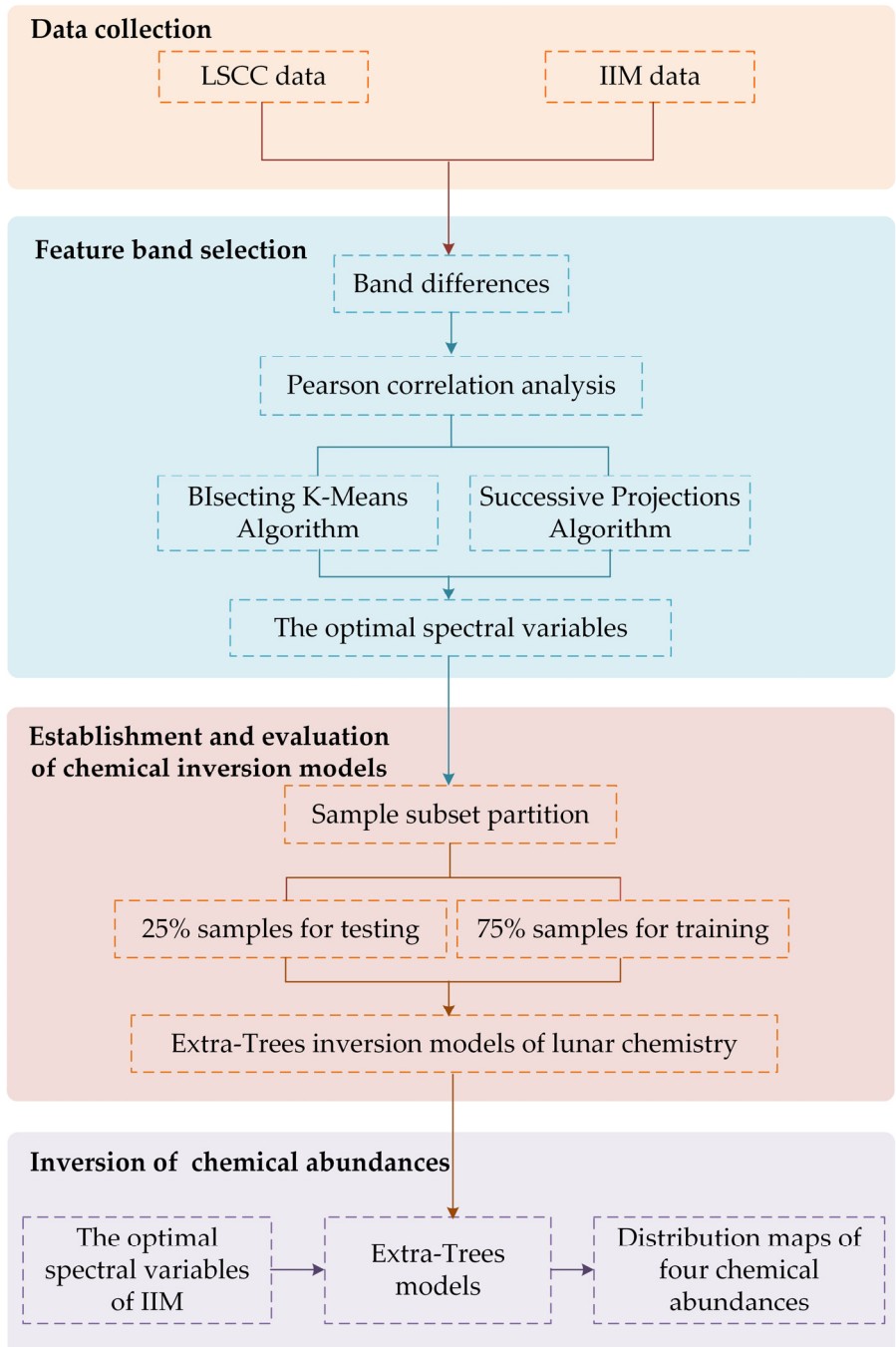

**Figure 1.** Flow chart of lunar surface chemical inversion.

*2.1. Data Description*

2.1.1. LSCC Data

The actual measured samples are important for studying the material components of the lunar surface and are the only standard for testing the effectiveness of the inversion. The LSCC soil sample data are stored in a spectral library created through the spectral analysis of samples collected from the six Apollo lunar landings. The samples include 10 samples of highland lunar soil from Apollo 14 and Apollo 16 and 9 samples of mare lunar soil from Apollo 11, 12, 15, and 17 [42]. Since the lunar surface spectral characteristics are influenced by the particle size of the material, the LSCC screened the measured samples for particle size and divided them into four groups, namely, <10 μm, 10–20 μm, 20–45 μm, and <45 μm, for a total of 76 subsamples [27,42–44]. All measured spectra of lunar surface

samples were obtained at the RELAB laboratory at Brown University using a phase angle of 30° at a sampling interval of 5 nm in the spectral range of 0.3–2.6 μm [40–42]. Composition data for the LSCC soils used in this paper are shown in Table 1.

**Table 1.** Composition data for LSCC soils used in this paper.

| Region | Mission | Number | Size (μm) | FeO | TiO$_2$ | Al$_2$O$_3$ | SiO$_2$ |
|---|---|---|---|---|---|---|---|
| mare | Apollo 11 | 10,084 | <10 | 12.00 | 7.25 | 15.90 | 42.10 |
| | | | 10–20 | 14.70 | 7.94 | 13.20 | 41.20 |
| | | | 20–45 | 15.50 | 8.30 | 12.00 | 41.30 |
| | Apollo 12 | 12,001 | <10 | 12.50 | 2.78 | 14.90 | 46.00 |
| | | | 10–20 | 15.90 | 2.96 | 12.30 | 45.00 |
| | | | 20–45 | 16.90 | 3.20 | 11.00 | 45.30 |
| | | 12,030 | <10 | 14.30 | 3.01 | 13.90 | 46.20 |
| | | | 10–20 | 17.20 | 3.32 | 10.70 | 46.30 |
| | | | 20–45 | 17.60 | 3.74 | 10.50 | 46.10 |
| | Apollo 15 | 15,041 | <10 | 11.00 | 1.79 | 16.40 | 46.60 |
| | | | 10–20 | 14.40 | 1.88 | 13.50 | 46.20 |
| | | | 20–45 | 15.20 | 2.03 | 12.50 | 46.10 |
| | | 15,071 | <10 | 9.59 | 1.57 | 17.10 | 46.90 |
| | | | 10–20 | 15.40 | 1.88 | 12.90 | 45.70 |
| | | | 20–45 | 15.60 | 2.33 | 12.40 | 45.80 |
| | Apollo 17 | 70,181 | <10 | 12.70 | 6.54 | 15.40 | 41.50 |
| | | | 10–20 | 15.50 | 7.88 | 12.70 | 40.40 |
| | | | 20–45 | 16.00 | 8.11 | 11.50 | 40.70 |
| | | 71,061 | <10 | 14.80 | 7.89 | 13.80 | 40.20 |
| | | | 10–20 | 17.50 | 8.94 | 10.80 | 39.50 |
| | | | 20–45 | 18.50 | 9.48 | 9.33 | 39.20 |
| | | 71,501 | <10 | 13.50 | 8.27 | 14.50 | 40.40 |
| | | | 10–20 | 16.40 | 9.83 | 11.60 | 39.00 |
| | | | 20–45 | 17.80 | 10.70 | 9.94 | 38.40 |
| | | 79,221 | <10 | 11.30 | 5.83 | 15.90 | 42.30 |
| | | | 10–20 | 15.00 | 7.21 | 12.90 | 40.90 |
| | | | 20–45 | 15.80 | 7.38 | 11.60 | 40.50 |
| highland | Apollo 14 | 14,141 | <10 | 7.66 | 1.51 | 19.20 | 49.20 |
| | | | 10–20 | 9.46 | 1.71 | 17.20 | 48.40 |
| | | | 20–45 | 11.60 | 1.96 | 15.00 | 47.20 |
| | | 14,163 | <10 | 8.83 | 2.07 | 18.90 | 47.20 |
| | | | 10–20 | 10.10 | 1.88 | 17.00 | 47.40 |
| | | | 20–45 | 11.50 | 2.00 | 15.40 | 47.10 |
| | | 14,259 | <10 | 7.82 | 2.02 | 19.30 | 47.90 |
| | | | 10–20 | 9.71 | 1.96 | 17.40 | 47.50 |
| | | | 20–45 | 11.00 | 1.99 | 15.80 | 47.10 |
| | | 14,260 | <10 | 8.10 | 1.94 | 19.10 | 47.80 |
| | | | 10–20 | 9.84 | 1.98 | 17.30 | 47.50 |
| | | | 20–45 | 10.70 | 1.86 | 16.30 | 47.40 |
| | Apollo 16 | 61,221 | <10 | 3.64 | 0.50 | 28.50 | 44.50 |
| | | | 10–20 | 4.40 | 0.54 | 27.50 | 44.50 |
| | | | 20–45 | 4.62 | 0.56 | 27.20 | 44.50 |
| | | 61,141 | <10 | 3.66 | 0.59 | 27.40 | 44.90 |
| | | | 10–20 | 5.14 | 0.64 | 25.60 | 44.60 |
| | | | 20–45 | 5.15 | 0.58 | 26.10 | 44.50 |
| | | 62,231 | <10 | 3.63 | 0.58 | 27.40 | 45.00 |
| | | | 10–20 | 4.86 | 0.61 | 26.30 | 44.70 |
| | | | 20–45 | 5.31 | 0.58 | 25.70 | 44.50 |
| | | 64,801 | <10 | 3.84 | 0.61 | 27.70 | 44.80 |
| | | | 10–20 | 4.78 | 0.68 | 26.30 | 44.50 |
| | | | 20–45 | 4.82 | 0.63 | 26.50 | 44.60 |
| | | 67,461 | <10 | 3.35 | 0.34 | 29.40 | 44.50 |
| | | | 10–20 | 4.64 | 0.39 | 27.80 | 44.10 |
| | | | 20–45 | 4.93 | 0.44 | 27.30 | 44.40 |
| | | 67,481 | <10 | 3.61 | 0.42 | 29.10 | 44.50 |
| | | | 10–20 | 4.04 | 0.40 | 28.40 | 44.40 |
| | | | 20–45 | 5.19 | 0.49 | 26.70 | 44.70 |

### 2.1.2. IIM Data

The IIM, one of the payloads of Chang'E-1, has achieved the first international application of interferometric imaging spectroscopy in the field of planetary exploration. The

wavelength range of IIM data is 480–960 nm, with a spectral resolution of 335 cm$^{-1}$, a total of 32 bands, and a spatial resolution of 200 m/pixel [18,45,46]. Critical to establishing a statistical relationship between the hyperspectral band characteristics of the samples and the mineral element content is the requirement that the laboratory-measured LSCC samples and the IIM image absolute calibrations have the same photometric calibration [7,21,33,47,48]. The IIM 2C data downloaded from the Ground Application System of the Lunar Exploration Project have been preprocessed with radiometric correction, geometric correction, and photometric correction [46,49]. The IIM data used in this paper are based on the processing of 2C data with further recalibration processing, which includes reflectance correction using Apollo samples. Therefore, the photometric conditions between the IIM image and the LSCC sample data are the same after preprocessing, and the laboratory-measured LSCC samples can be used together with the IIM data.

IIM data quality evaluation is also an important prerequisite for performing scientific inversion. As indicated by signal-to-noise ratio analysis, the first four bands of IIM 2C data contain considerable noise, and the 32nd band has the highest noise level (lowest signal-to-noise ratio observed). Therefore, 26 bands of data in the range between 513 and 891 nm (bands 5–30) are used in this paper.

### 2.2. Feature Band Selection

The reflectance spectra used in this paper are between 513 and 891 nm, and band difference calculation results in 325 spectral bands. If all these spectral bands are input into the model for prediction, the number of bands will be high, as will the correlations between adjacent bands, which will inevitably lead to an increase in the redundancy of spectral information and adversely affect the accuracy and generalization of the model [37]. Therefore, it is necessary to filter the sensitive bands that play a key role in the model.

Pearson correlation analysis is used to correlate the differentially transformed spectral features and the oxide content to find the band differences with correlation coefficients that pass the significance test at the 0.01 level. The results are used to identify the sensitive feature regions of the differentially transformed spectra.

Due to the large number and continuous distribution of selected wavelengths, secondary screening is required within the sensitive regions identified with Pearson correlation analysis. Considering the random nature of the initial bands of the SPA, to reduce the possibility of the selection of invalid initial bands, the range of random initial bands is restricted to the corresponding clusters by combining the above algorithm with a clustering algorithm. Fine screening is performed separately in each cluster to eliminate the presence of covariance bands. The feature wavelengths selected by the SPA are those that are most representative.

#### 2.2.1. Bisecting K-Means Algorithm

The bisecting K-means algorithm is an improvement on and expansion of the K-means algorithm, and it essentially bifurcates the selected clusters until the specified number of clusters is reached while satisfying a criterion based on the sum of the squared error (SSE) [50]. The SSE is defined as shown in Equation (1). The greatest advantage of this algorithm over the traditional K-means method is that it is simple and fast to implement and finds the globally optimal solution. Therefore, in this paper, the dichotomous K-means algorithm is used for the clustering analysis of sensitive feature regions with differences, and the initial K value is set to 3.

$$SSE = \sum_{i=1}^{k} \sum_{x \in C_i} dist(c_i, x)^2 \tag{1}$$

where $k$ is the number of clusters, $c_i$ is the cluster center of cluster $C_i$, and $x$ is a sample in that cluster.

2.2.2. Successive Projections Algorithm

The successive projections algorithm (SPA) is a forward iterative feature variable selection method that minimizes the covariance in vector space, minimizes the redundant information in the original spectral matrix, eliminates the effect of covariance, and can be used for spectral feature wavelength screening [51,52]. The SPA starts with one wavelength and introduces the wavelength with the largest projection vector into the set of wavelength variables in each iteration by analyzing the projection of the vector until a specified number of wavelengths is reached [53–55]. The exact procedure of the algorithm is as follows.

1.　In the 1st iteration ($n = 1$), any column of wavelength $k(0)$ is chosen and denoted as $x_j$, where $j = 1, \ldots, J$.
2.　The wavelengths that are not included in the set are identified as $s = \{j, 1 \leq j \leq J, j \notin \{k(0), \ldots, k(n-1)\}\}$.
3.　The projection of the initialized band $x_j$ with an unselected wavelength in orthogonal space is calculated as

$$P_{x_j} = x_j - \left(x_j^T x_{k(n-1)}\right) x_{k(n-1)} \left(x_{k(n-1)}^T x_{k(n-1)}\right)^{-1} \tag{2}$$

4.　The maximum wavelength of the projection vector is calculated:

$$k(n) = \arg\left(\max\left(\|P_{x_j}\|\right)\right) \tag{3}$$

5.　$n = n + 1$; if $n < N$, return to step 2.
6.　The final combination of wavelength variables is determined.

*2.3. Sample Subset Partition*

The division of the sample data set will affect the accuracy of the model estimates to some extent. The SPXY algorithm was proposed by Galvao et al. [56]. The objective is to use the physical-chemical variable $y$ and the spectral variable $x$ to calculate the intersample distance, fully characterize the sample distribution, effectively cover the multidimensional vector space, increase intersample variability and representativeness, and improve model stability [57,58]. The distance equation is as follows [56]:

$$d_x(p,q) = \sqrt{\sum_{j=1}^{J} \left[x_p(j) - x_q(j)\right]^2} \ p,q \in [1,N] \tag{4}$$

$$d_y(p,q) = \sqrt{\left(y_p - y_q\right)^2} = |y_p - y_q| \ p,q \in [1,N] \tag{5}$$

To ensure that the variables $x$ and $y$ give the same weight to a sample, $d_x(p,q)$ and $d_y(p,q)$ are divided by the maximum value in the data set, respectively. Thus, the formula after normalization is

$$d_{xy}(p,q) = \frac{d_x(p,q)}{max_{p,q\in[1,N]}d_x(p,q)} + \frac{d_y(p,q)}{max_{p,q\in[1,N]}d_y(p,q)} \ p,q \in [1,N] \tag{6}$$

In this study, the SPXY algorithm was used to divide the training and test sets of 76 LSCC samples based on the four oxide contents as $y$ variables and the spectral features as $x$ variables, and the specific division results are shown in Table 2. The training set accounts for 75% of all samples and contains 57 samples, and the test set accounts for 25% of all samples and contains 19 samples. The statistical parameters of the oxide content in the test set are generally within the same ranges as those of the training set, and the sample set is divided reasonably and effectively, which helps improve the stability and reliability of the model.

**Table 2.** Statistical parameters of sample set partitioning.

| Elements | Training Set | | | | | Test Set | | | | |
| | Count | Max | Min | Mean | Std | Count | Max | Min | Mean | Std |
|---|---|---|---|---|---|---|---|---|---|---|
| FeO | 57 | 18.50 | 3.35 | 10.50 | 5.01 | 19 | 16.40 | 3.66 | 10.83 | 4.40 |
| TiO$_2$ | 57 | 10.70 | 0.34 | 3.49 | 3.21 | 19 | 7.94 | 0.35 | 2.38 | 2.41 |
| Al$_2$O$_3$ | 57 | 29.40 | 9.33 | 18.90 | 6.83 | 19 | 27.40 | 11.60 | 16.52 | 4.96 |
| SiO$_2$ | 57 | 49.20 | 38.40 | 44.35 | 2.75 | 19 | 47.60 | 40.40 | 44.90 | 2.31 |

*2.4. Extra-Trees Regression*

Extra-Trees is an integrated algorithm based on decision trees with good generalization and robustness and was proposed by Geurts et al. [38]. The Extra-Trees algorithm is trained using all training samples, and different decision trees are constructed according to different features. The score of each random classification node for K random features is calculated with the Score function (Equation (7)), and the node with the highest score is selected as the splitting node [59,60]. When a new sample is input, multiple decision trees in the integrated model score it, and the final prediction is based on the average of all decision tree predictions [61].

$$Score_R(s, S) = \frac{var\{y|S\} - \frac{|S_l|}{|S|} var\{y|S_l\} - \frac{|S_r|}{|S|} var\{y|S_r\}}{var\{y|S\}} \tag{7}$$

where $var\{y|S\}$ is the variance of the output $y$ in sample $S$ and $l$ and $r$ denote the left and right divergence values of the nodes, respectively.

Unlike random forests that build each tree by sampling with feedback, Extra-Trees uses the entire training sample to establish each tree, which can effectively reduce the effects of the bias and variance of the sample set [62]. Moreover, unlike random forests that obtain the best bifurcation within a random subset, Extra-Trees obtains the bifurcation of each decision tree completely randomly [38].

In this paper, the Extra-Trees algorithm is used as the lunar surface oxide content prediction model, and the flow chart of this model is shown in Figure 2. The best combination of bands after secondary screening is input into the model, and each oxide content is output to construct the Extra-Trees oxide content prediction model. The implementation of the Extra-Trees model in this study is based on the ExtraTreesRegressor algorithm provided in the sklearn package of the Python language. The algorithm has two main parameters, including the minimum size of the samples for splitting the nodes, $n_{min}$, and the size of the randomly selected attributes for each node, $k$ [38]. According to Geurts's experimental results, the default parameter settings are generally satisfactory in terms of accuracy and computational efficiency; therefore, the default parameters are used.

*2.5. Evaluation Indicators*

The accuracy and predictive ability of the model are mainly evaluated in terms of the coefficient of determination ($R^2$) and root mean square error (RMSE). $R^2$ is used to evaluate the correlation between the predicted and true values of the sample, and the closer $R^2$ is to 1, the higher the correlation between the predicted and true values (Equation (8). The RMSE is used to evaluate the predictive ability of the model for a given data set, and the smaller the RMSE is, the better the predictive ability of the model [37,52].

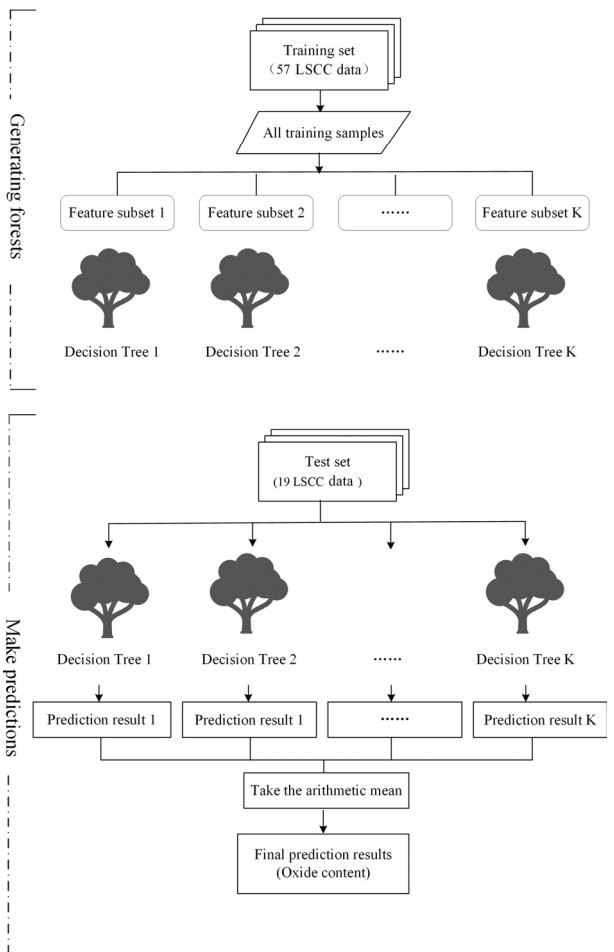

**Figure 2.** Flow chart of the Extra-Trees model.

$$R^2 = 1 - \frac{\sum_{i=1}^{m}(\check{c}_i - c_i)^2}{\sum_{i=1}^{m}(c_i - \overline{c_i})^2} \tag{8}$$

$$RMSE = \sqrt{\frac{\sum_{i=1}^{m}(\check{c}_i - c_i)^2}{n}} \tag{9}$$

where $\check{c}_i$, $c_i$ and $\overline{c_i}$ denote the true, predicted, and mean values of the $i$th sample, respectively, and $n$ denotes the number of samples.

## 3. Results

### 3.1. Correlation Coefficients between Elements and Reflectance

In this paper, IIM reflectivity data in the range of 513–891 nm were used as the basis for the estimation and analysis of the lunar surface oxide content. First, the standard bidirectional emissivity data for 76 LSCC samples (300–2600 nm) were resampled into the wavelength range of IIM data (513–891 nm) using a linear interpolation method. The spectral data after resampling were differentially transformed to obtain the reflectivity spectral band difference.

The linear correlation coefficients between LSCC reflectance and the content were calculated (Figure 3). The results showed that the correlation coefficients did not vary significantly within 513–891 nm, and the correlation coefficients between FeO, $TiO_2$, $Al_2O_3$, and $SiO_2$ and reflectance remained at approximately 0.75, 0.58, 0.79, and 0.25, respectively. Additionally, positive correlations were observed between the $Al_2O_3$ and $SiO_2$ contents

and spectral reflectance, and negative correlations were observed between the FeO and $TiO_2$ contents and spectral reflectance.

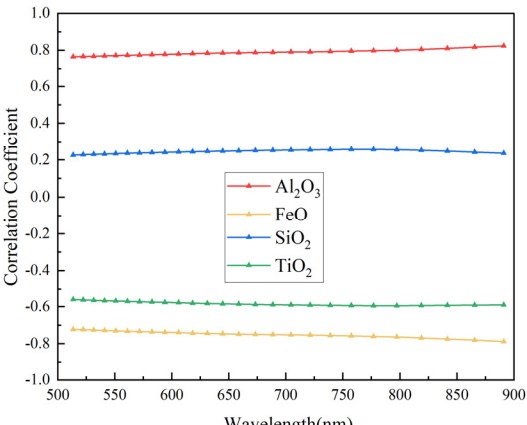

**Figure 3.** Linear correlation coefficients between the contents of the four oxides and LSCC reflectance data.

The correlations between the elemental contents and LSCC reflectance are mainly concentrated in the grain size range of 10–20 μm, as the optical information at these sizes is most similar to that of the bulk soil [18,63]. Figure 4 demonstrates the relationship between the 891 nm reflectance and the content of each oxide (FeO and $TiO_2$) in the LSCC samples of size 10–20 μm. The FeO and $TiO_2$ contents decrease with increasing reflectance, highlighting the negative correlations between the FeO and $TiO_2$ contents and spectral reflectance. However, the relationship between elements and IIM reflectance is not necessarily linear. For some oxides, it is difficult to describe the complex relationship between them and the spectra based on conventional linear regression [36], and suitable nonlinear models are needed to invert the major oxide contents.

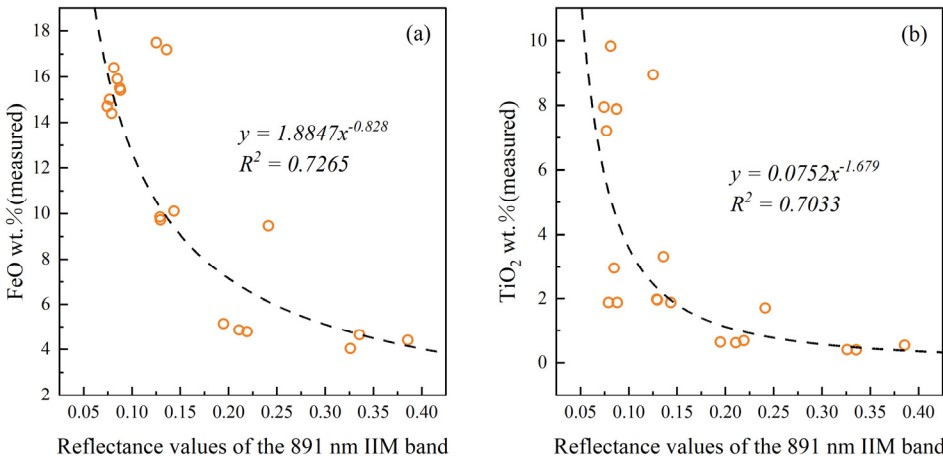

**Figure 4.** Relationship between the 891 nm reflectance and the (**a**) FeO and (**b**) $TiO_2$ contents in LSCC samples with 10–20 μm grain sizes.

Moreover, there are also correlations among elements. As shown in Figure 5, there is a significant inverse correlation between the FeO and $Al_2O_3$ contents and a significant inverse correlation between $TiO_2$ and $SiO_2$. These findings are consistent with the conclusion that there are positive correlations between the $Al_2O_3$ and $SiO_2$ contents and spectral reflectance and negative correlations between the FeO and $TiO_2$ contents and spectral reflectance, as shown in Figure 3.

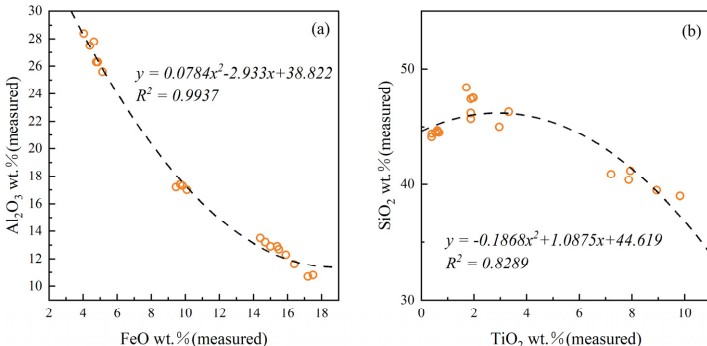

**Figure 5.** Correlations between (**a**) FeO and Al$_2$O$_3$ and (**b**) TiO$_2$ and SiO$_2$ in the LSCC samples with 10–20 μm grain sizes.

Figure 6 shows the correlations between the reflectance band differences and the major oxides, and the color of each point in the figure indicates the magnitude of the correlation coefficient between the band differences and the oxide contents. The results show that the band differences display the highest correlations with FeO and Al$_2$O$_3$ and the lowest correlations with TiO$_2$ and SiO$_2$. The maximum value of the correlation coefficient between FeO and the band difference is located at 541 nm–532 nm (0.861); most of the correlation coefficients between Al$_2$O$_3$ and the band difference of IIM are approximately 0.80, and the maximum value is 0.870. Additionally, the correlation between TiO$_2$ and the band difference of IIM decreases from the lower left to the upper right corner, and the maximum value appears at 532 nm–522 nm (0.710). The correlation between SiO$_2$ and the band difference is poor, with a maximum value of 0.427, and most of the correlation coefficients are approximately 0.40. In general, for the four oxides, the high correlations are mainly concentrated in the lower left corner, and the band differences among the longer bands (upper right corner in the figure) are poorly correlated. Correlation calculations for the original bands and all band differences in the IIM spectral range separately show that band differences can be used to effectively enhance the correlations between reflectance and the oxide contents to determine the best combination of bands for subsequent modeling.

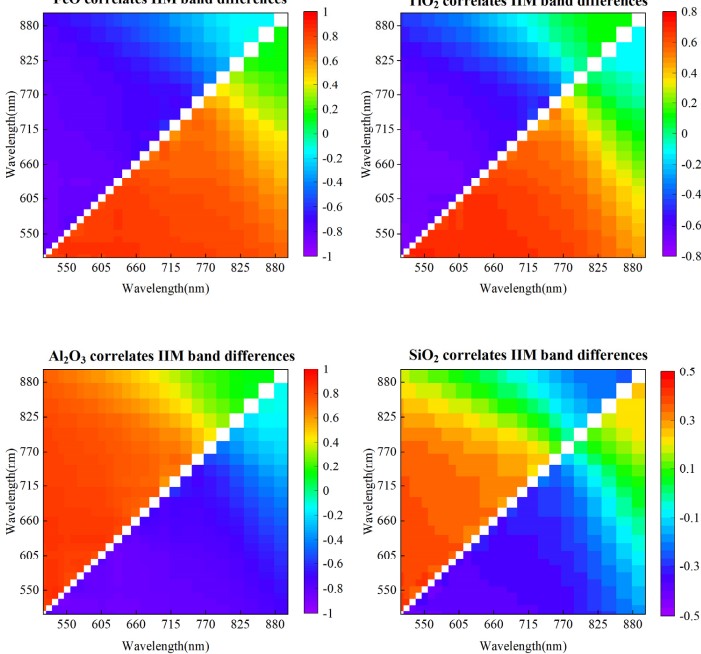

**Figure 6.** Correlation coefficients between spectral differences and chemical abundances. Each point represents the band difference between the x− and y−axes, and the intensity of each point is the correlation between the band difference and chemical abundance.

### 3.2. Screening of Feature Bands

The band differences that passed the 0.01 significance test were screened twice, and the sensitive bands identified through BKM clustering analysis combined with SPA are shown in Table 3. These bands are used as the inputs to the model.

**Table 3.** Final screening results for feature bands.

| Oxide | Clustering Categories | Number of Bands | Selected Band (nm) |
|---|---|---|---|
| FeO | Category 1 | 5/91 | 522–583, 631–777, 645–721, and 659–842, and 659–891 |
| | Category 2 | 9/123 | 513–757, 513–819, 522–659, 551–721, 551–777, 561–842, 606–777, 606–891, and 618–819 |
| | Category 3 | 3/99 | 532–583, 739–819, and 739–891 |
| TiO$_2$ | Category 1 | 1/81 | 513–631 |
| | Category 2 | 1/119 | 522–673 |
| | Category 3 | 3/87 | 606–659, 721–819, and 739–797 |
| Al$_2$O$_3$ | Category 1 | 1/91 | 522–631 |
| | Category 2 | 4/123 | 513–891, 561–797, 606–757, and 645–842 |
| | Category 3 | 3/98 | 513–561, 739–891, and 739–819 |
| SiO$_2$ | Category 1 | 10/125 | 513–618, 522–594, 532–583, 541–572, 551–645, 561–673, 572–689, 583–631, 606–659, and 631–757 |
| | Category 2 | 12/46 | 513–631, 513–673, 522–659, 522–689, 532–645, 551–721, 561–739, 572–721, 572–757, 583–777, 606–739, and 618–777 |
| | Category 3 | 8/32 | 513–721, 513–777, 522–705, 522–819, 532–739, 551–757, 561–777, and 583–797 |

### 3.3. Establishment and Evaluation of the Extra-Trees Model

The spectral features obtained based on the band reduction screening were used as the independent variables for the Extra-Trees modeling analysis. Additionally, the lunar surface oxide content was used as the dependent variable, and the SPXY algorithm was applied to divide the sample set and establish the Extra-Trees model. To effectively analyze the results of variable screening, the modeling results for the original bands were also applied for comparison. The prediction results (Figures 7 and 8) show that the coefficients of determination R$^2$ of the FeO, TiO$_2$, Al$_2$O$_3$, and SiO$_2$ prediction models after differential transformation and variable filtering were 0.962, 0.944, 0.964, and 0.860, respectively, which are all greater than 0.850, indicating good prediction ability for the four oxide contents. The model accuracy is improved compared with that obtained based on the original bands, further verifying the importance of the differential transformation of the original bands and reduced-dimension filtering of the variable information. The modeling accuracy of feature band selection was compared with the accuracy reported by Wu [18] (Table 4), and the accuracy for all four oxide contents after differential transformation and variable screening was improved. Therefore, in this paper, the differential variables obtained based on the SPA were adopted as the inputs to the model for prediction.

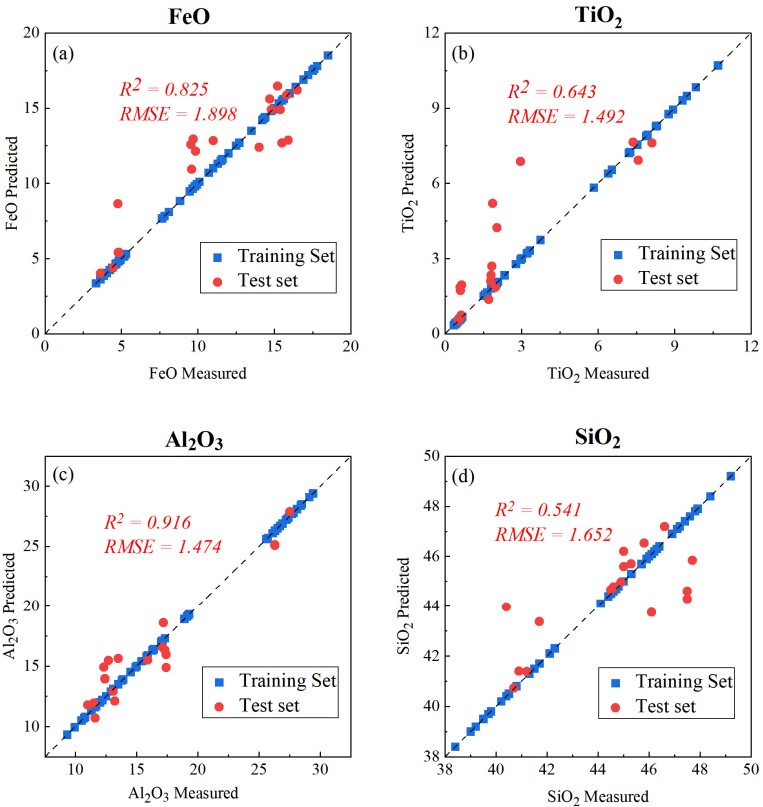

**Figure 7.** Training and testing accuracy of the Extra-Trees model with chemical abundance based on the original waveband: (**a**) FeO, (**b**) $TiO_2$, (**c**) $Al_2O_3$, and (**d**) $SiO_2$.

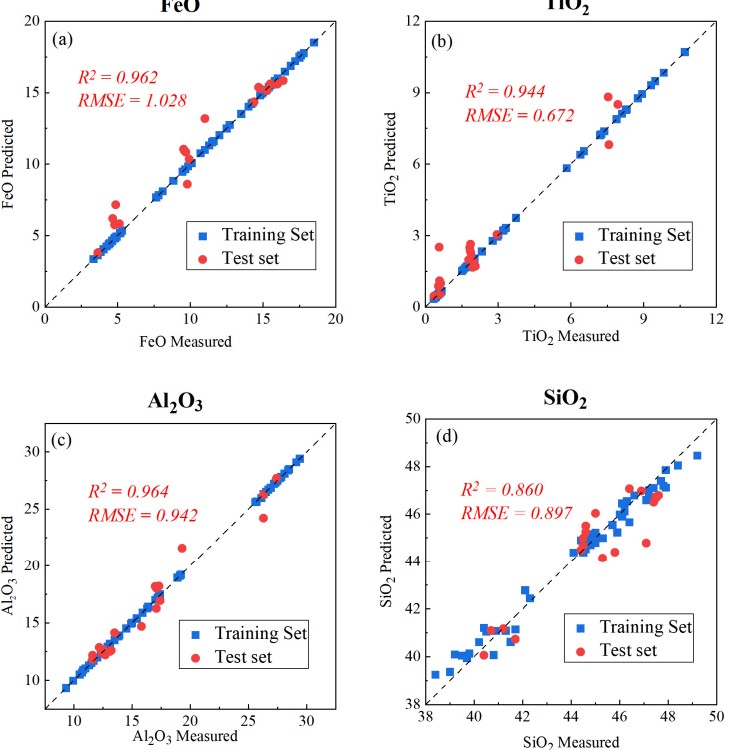

**Figure 8.** Training and testing accuracy of the Extra-Trees model with chemical abundance based on feature band selection: (**a**) FeO, (**b**) $TiO_2$, (**c**) $Al_2O_3$, and (**d**) $SiO_2$.

**Table 4.** Comparison of the accuracy of the four chemical abundance tests with the results of Wu [18].

| | FeO (wt.%) | TiO$_2$ (wt.%) | Al$_2$O$_3$ (wt.%) | SiO$_2$ (wt.%) |
|---|---|---|---|---|
| Calibration in this work (Extra-Trees model based on feature band selection) | | | | |
| R$^2$ | 1 | 1 | 1 | 0.974 |
| RMSE | 0.012 | 0 | 0.032 | 0.438 |
| Validation in this work (Extra-Trees model based on feature band selection) | | | | |
| R$^2$ | 0.962 | 0.944 | 0.964 | 0.860 |
| RMSE | 1.028 | 0.672 | 0.942 | 0.897 |
| Calibration by Wu [18] | | | | |
| R$^2$ | 0.90 | 0.69 | 0.92 | 0.76 |
| RMSE | 1.58 | 2.00 | 1.68 | 0.75 |
| Validation by Wu [18] | | | | |
| R$^2$ | 0.88 | 0.59 | 0.90 | 0.67 |
| RMSE | 1.76 | 2.26 | 1.92 | 0.91 |

### 3.4. Extra-Trees Modelling in the Apollo 17 Area

The Apollo 17 area is close to the junction of Serenitatis and Tranquillitatis, and the coordinates of the landing site are (30°44′58.3″E, 20°9′50.5″N). The geological environment of the area is complex; according to the analysis of Apollo17 samples, the rock types in the area are mainly overlying mare basalts and orange-grey breccia [64].

#### 3.4.1. Extra-Trees Modelling of FeO

Fe mainly exists in lunar mafic minerals, and understanding the abundance and distribution of Fe can aid in understanding the nature and origin of the Moon and provide an indicator of lithology to distinguish between mare and highland [65]. In this paper, Extra-Trees modeling is performed with the effective bands obtained by the Pearson correlation coefficients and SPA. Due to the large number of bands of IIM data that undergo band-difference transformation, the coverages overlap. The sensitive bands that play a key role in the model are first screened, and the model is then trained and tested; however, the regression results are not the only reference used to evaluate the model. To further verify the accuracy of the model, it was applied to the Apollo 17 area (landing site coordinates: 30°44′58.3″ E, 20°9′50.5″ N) to evaluate the accuracy of FeO inversion.

The FeO training and testing accuracies for the Extra-Trees model after variable screening are shown in Figure 8a, where the blue dots represent the training set, and the red dots represent the test set. The coefficient of determination (R$^2$) for the test set is 0.962, and the RMSE is approximately 1.028. The FeO inversion results (Figure 9) show that the predicted FeO abundance in the region near the Apollo 17 landing site ranges from 7.64 to 15.72 wt.%, and the FeO content of the mare area is significantly higher than that of the highland area, with good consistency regarding the distribution of FeO content values in comparison with the results of Lucey et al. [17].

#### 3.4.2. Extra-Trees Modelling of TiO$_2$

The distribution of the TiO$_2$ content varies greatly among different types of mare basalts; thus, it is the key to classifying mare basalt types and is important for the exploitation of lunar ilmenite resources [6]. The optimal band selection for TiO$_2$ is similar to that in the FeO model. First, the 287 bands that passed the $p < 0.01$ significance test were used to establish the sensitive region for TiO$_2$, and the best band combination was selected by clustering analysis and applying the SPA. The results are shown in Table 4.

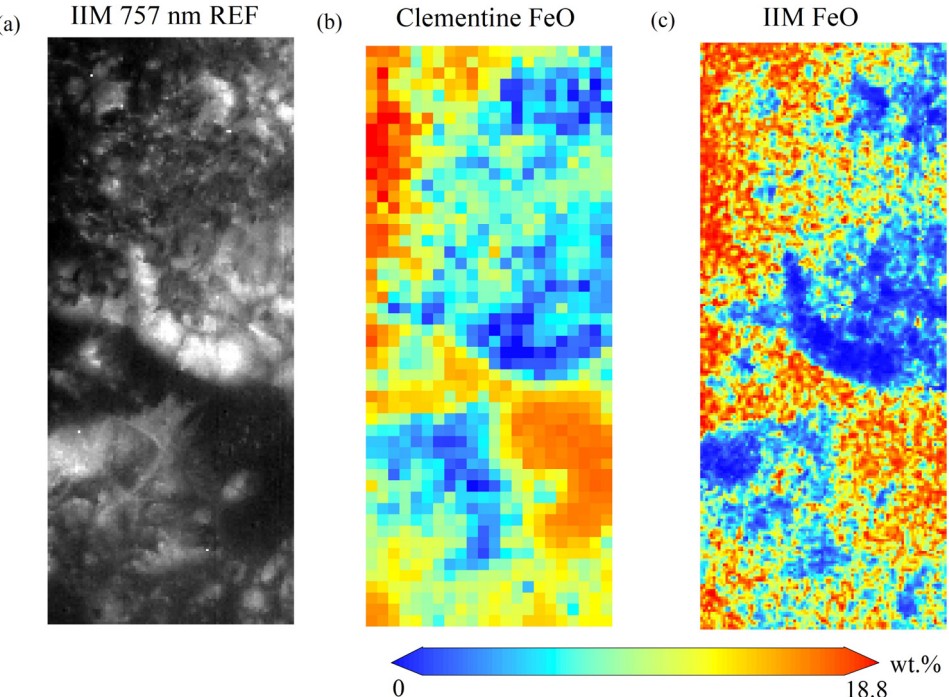

**Figure 9.** Apollo 17 area: (**a**) IIM 757 nm reflectance image, (**b**) FeO contents from Lucey et al. [17], and (**c**) FeO prediction results of the Extra-Trees model based on feature band selection.

The $TiO_2$ training and testing accuracies of the Extra-Trees model after variable screening are shown in Figure 8b, where the coefficient of determination ($R^2$) for the test set is 0.944 and the RMSE is approximately 0.672. The model was applied for $TiO_2$ inversion in the test area (Figure 10), and the results show that the $TiO_2$ abundance ranges from 2.21 to 9.37 wt.%; the high-value areas are mainly located in the mare, and the low-value areas are mainly in the highland, which is generally similar to the results reported by Lucey et al. [17]. The inversion results display a similar trend, but the lower limit value of $TiO_2$ is higher, and the contents are slightly higher in the lower left corner of the figure, which may be due to the effect of topographic shadowing from the light angle.

3.4.3. Extra-Trees Modelling of $Al_2O_3$

Almost all Al on the lunar surface is present in plagioclase, and the distribution of the $Al_2O_3$ content is related to the formation and evolution of the lunar crust; notably, levels are highest in highland calcarenite rocks and intermediate in mare basalts and can be used to assist in distinguishing mare basalts from highland plagioclase. To assess the accuracy of its inversion, an initial screening based on Pearson correlation coefficients was carried out, followed by BKM clustering analysis combined with the SPA algorithm to find the best band combinations for $Al_2O_3$. From Table 3, it can be seen that the eight best band differences were finally screened and used as input to the model for modeling.

The training and testing accuracies of the $Al_2O_3$ Extra-Trees model after variable screening are shown in Figure 8c. The coefficient of determination ($R^2$) for the test set with this method is 0.964, and the RMSE is approximately 0.942. The model was applied to the Apollo 17 region for $Al_2O_3$ inversion, and the results (Figure 11) show that the $Al_2O_3$ abundance obtained with this method ranges from 11.75–24.43 wt.%, with a mean value of 14.70 wt.%. The abundance of $Al_2O_3$ in the highlands region is lower than that in the mare region, and this correlates negatively with FeO.

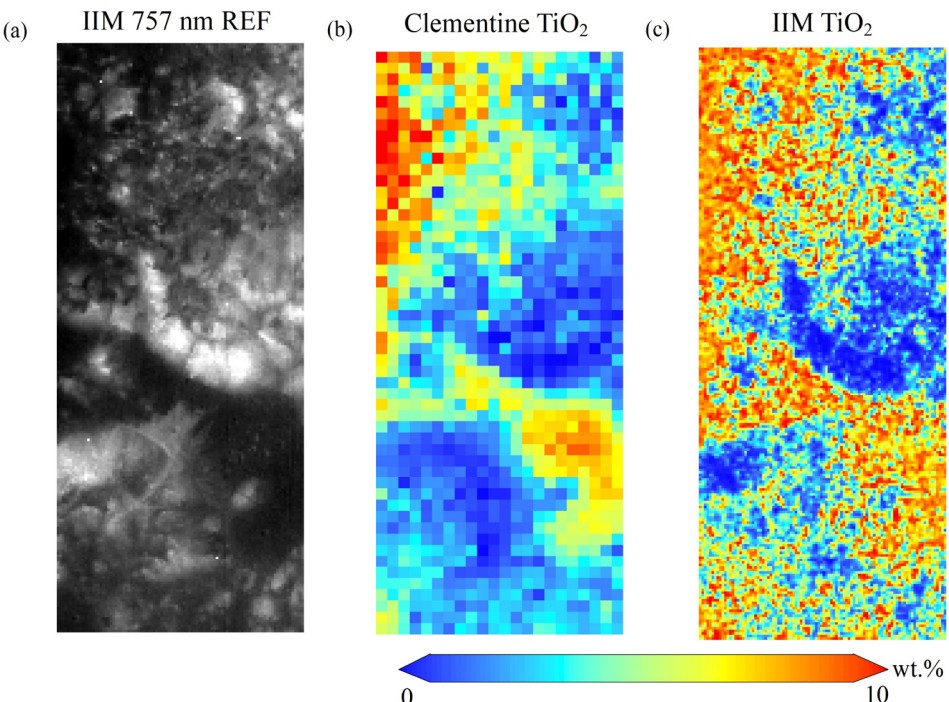

**Figure 10.** Apollo 17 area: (**a**) IIM 757 nm reflectance image, (**b**) $TiO_2$ contents from Lucey et al. [17], and (**c**) $TiO_2$ prediction results of the Extra-Trees model based on feature band selection.

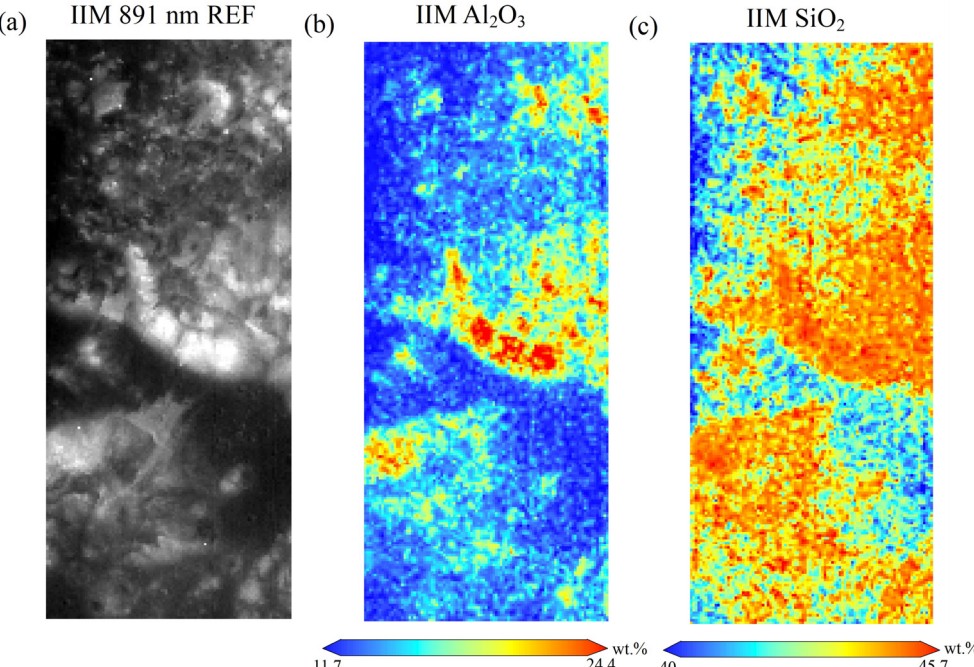

**Figure 11.** Apollo 17 area: (**a**) IIM 891 nm reflectance image, (**b**) $Al_2O_3$, (**c**) $SiO_2$ prediction results of the Extra-Trees model based on feature band selection.

### 3.4.4. Extra-Trees Modeling of $SiO_2$

$SiO_2$ is in nearly every rock and mineral grain on the Moon as the fundamental component of silicate minerals. The optimal waveband selection for $SiO_2$ is similar to that for FeO, and the optimum number of bands for $SiO_2$ is shown in Table 3, with a total of 30 band differences.

The training and testing accuracies of the $SiO_2$ Extra-Trees model after variable screening are shown in Figure 8d. The $R^2$ value for the test set obtained with this method is 0.860, and the RMSE is approximately 0.897. The model was applied for $SiO_2$ inversion in the Apollo 17 region (Figure 11c), and the results show that the $SiO_2$ abundance obtained with this method ranges from 41.08 to 45.67 wt.%, with a mean value of 43.84 wt.%. $SiO_2$ is commonly found in lunar rocks, with higher levels in the highland than in the mare, which is negatively correlated with $TiO_2$.

### 3.5. Oxide Content Mapping for the Copernicus Crater Region

3.5.1. Regional Distribution of Lunar Surface Chemistry

Using ArcGIS as the platform, a 1:2,500,000 lunar geological map of the study area was mapped (Figure 12), which covers a geographical area of approximately $125 \times 10^4$ km$^2$ at longitudes of $-52°$ to $6.6°$ W and latitudes of $-2.8°$ to $28°$ N. The study area is mainly occupied by the Copernicus crater, Kepler crater, and Aristarchus crater, located in the transition zone between the mare and the lunar highlands in the south of the Oceanus Procellarum. The study area is rich in geomorphic features and material types, and the rocks are mainly mare basalts, followed by KREEP rocks and ferroan anorthositic suite. Linear structures are widely developed and distributed and are of great importance to studies of lunar diagenesis and geological evolution.

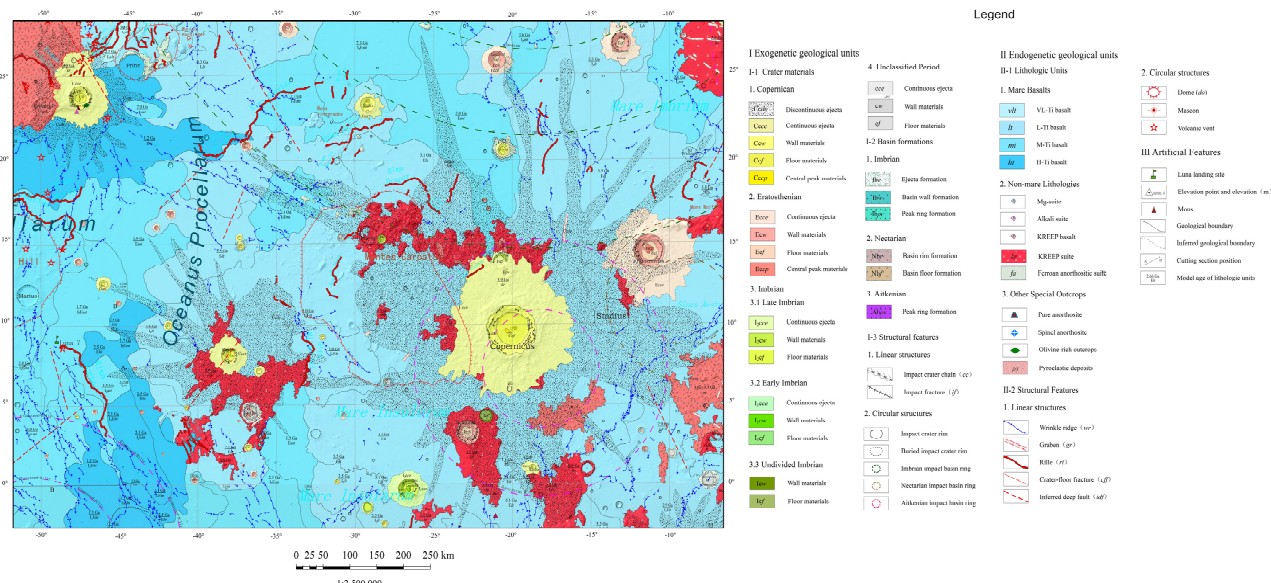

**Figure 12.** A 1:2,500,000 geological map [66] of the study area.

In the study area, the mare basalt is widely distributed in the lower and flat region, including south of Oceanus Procellarum and the impact basin floor, basically occupying the entire study area. These mare basalts formed between 19 Ga and 37 Ga ago according to the isotopic age data of lunar samples and the dating results of crater size-frequency distribution (CSFD) for mare basalt units. Meanwhile, a large number of tectonic developments are present, and the main linear formations are rills, wrinkle ridges, and crater-floor fractures, with wrinkle ridges being the most numerous, widespread, and characteristic linear structures, mainly within the mare basalt. Rills are also widespread and distributed around the wrinkle ridges; the circular structures are mainly domes, volcanic vents, craters, and impact basins.

The main geological evolution in the study area includes endodynamic geological evolution processes, such as differentiation of magma ocean, plutonic magmatism, and volcanism, and exodynamic geological evolution processes such as impact. During the Magma-Oceanian period, the primary ferroan anorthositic crust mainly formed, followed by the formation of the Copernicus-H basin in the Aitkenian, the Imbrain basin in the Imbrian, the

Timocharis crater in the Eratothenian, and the Copernicus, Kepler, and Aristarchus craters in the Copernican.

An Extra-Trees model based on variable screening was used to invert the oxide contents in the region near the Copernicus crater using IIM data, and frequency histograms were plotted based on the oxide contents (Figures 13 and 14).

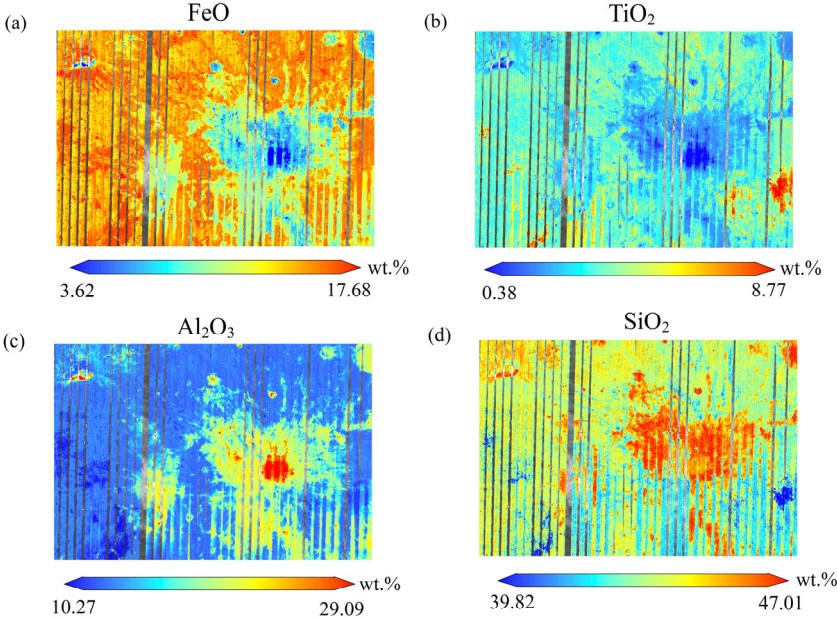

**Figure 13.** Predicted results of oxide contents in the study area: (**a**) FeO, (**b**) TiO$_2$, (**c**) Al$_2$O$_3$ and (**d**) SiO$_2$.

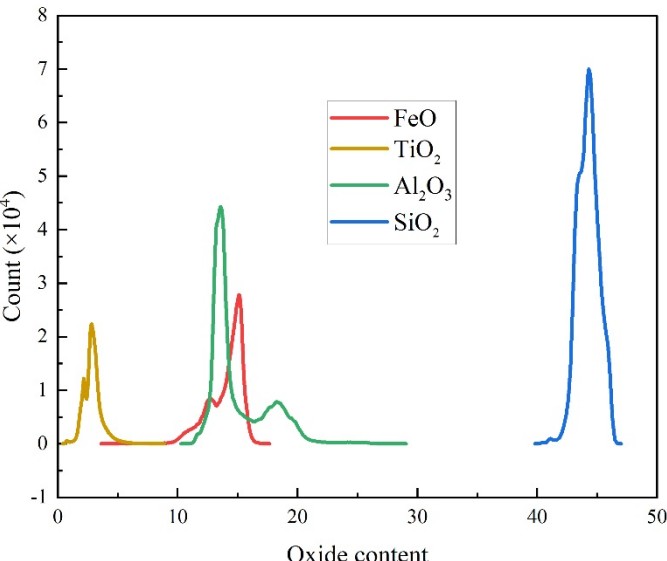

**Figure 14.** Smoothed histograms of four oxide contents in the study area.

Figure 13 shows the predicted results of oxide contents (FeO, TiO$_2$, Al$_2$O$_3$, and SiO$_2$) in the Copernicus crater region. As shown in Figure 13, the FeO content ranges from 3.61 to 17.68 wt.%, with a mean value of 12.51 wt.%. The FeO content in the mare is significantly higher than that in the highland, with the lower FeO content mainly located in the Copernicus crater, Aristarchus crater, and Kepler crater floor, and the high FeO content area mainly located on the southwest side of Kepler crater. The frequency histogram in Figure 14 also shows a clear bimodal distribution, with the first peak at approximately 12.80 wt.% and the second peak at approximately 15.1 wt.%. The TiO$_2$ content ranges from

0.38 to 8.88 wt.%, with a mean value of 2.60 wt.%. Its content is significantly higher in the mare than in the highland, and there is a positive correlation between the FeO content in areas with high $TiO_2$ content in the mare basalt area. The low $TiO_2$ content areas are mainly located around the Copernicus, Aristarchus, and Kepler craters, and the high $TiO_2$ content areas are mainly located on the southwestern side of the Kepler crater and the northeastern side of the Copernicus crater, while the rest of the mare has moderate $TiO_2$ content. The frequency histogram distribution shows a clear bimodal distribution, with the first peak at approximately 2.16 wt.% and the second peak at approximately 2.82 wt.%. The $Al_2O_3$ content ranges from 10.27–29.09 wt.%, with a mean value of 13.56 wt.%. It is clearly higher in the highland than in the mare, with an inverse correlation with FeO content. The high-value areas are mainly located in the vicinity of craters, with lower levels in the mare, especially on the southwestern and western sides of the Kepler crater. The frequency histogram of $Al_2O_3$ shows a clear bimodal distribution, with the first peak of approximately 13.57 wt.% and the second peak of approximately 18.25 wt.%. The $SiO_2$ content ranges from 39.82 to 47.01 wt.%, with an average value of 39.53 wt.%. The $SiO_2$ content is high throughout the study area, ranging from 39.82 to 47.01 wt.%, with a mean value of 39.53 wt.%. The content is significantly higher in the highland than in the mare, which shows an inverse correlation with $TiO_2$. The high-value areas are mainly located within the crater and its sputtering, with lower levels in the rest of the area. The $SiO_2$ first peak of the frequency histogram is approximately 41.11 wt.%, and the second peak is approximately 44.25 wt.%.

3.5.2. Comparison with Previous Works

Figures 15 and 16 show the FeO and $TiO_2$ content distributions generated based on the different data/methods. First, the correlation analysis was carried out with the FeO content distribution maps generated in this paper (Table 5). The correlations between the FeO inversion results in this paper and the inversion results of Clementine, IIM (band ratio), and LP GRS data were 0.91, 0.89, and 0.90, respectively, and the correlation was basically stable at approximately 0.90. The correlations between the $TiO_2$ inversion and the Clementine and IIM (band ratio) inversions (Table 6) were 0.61 and 0.65, respectively. Due to the low resolution of the GRS $TiO_2$ data, this correlation was not analyzed. As seen in Figure 15, the FeO and $TiO_2$ contents of the inversions in this paper are lower than the mean values of the inversions of other models. To better compare the discrepancy between the inversion results of this paper and the Clementine FeO results. the IIM FeO map is compared with the Clementine FeO product resampled to the same resolution (0.5°), and the correlation between the two products is 91%, with good agreement, as shown in Figure 17. The difference between the results of this paper and Clementine may be due to the influence of topographic shadows caused by the camera angle, post-image processing, and differences in the oxide inversion algorithm. Additionally, using IIM data, the difference between the results of this paper and the band ratio-based method is mainly due to differences in the oxide inversion algorithm and post-image processing. In contrast to the IIM optical image data, which can only capture the material component of the lunar surface, the LP GRS data are not affected by topographic shadows, light angles, etc., and can be detected to a depth of 20–30 cm below the lunar surface [15]. Therefore, the difference between the results of this paper and the LP GRS is mainly because the GRS data are not affected by topographic shadows and because the GRS data are deeper than the IIM data. The results from different data and methods indicate that, despite the different absolute abundance ranges, their relative abundances are similar to each other [19,33].

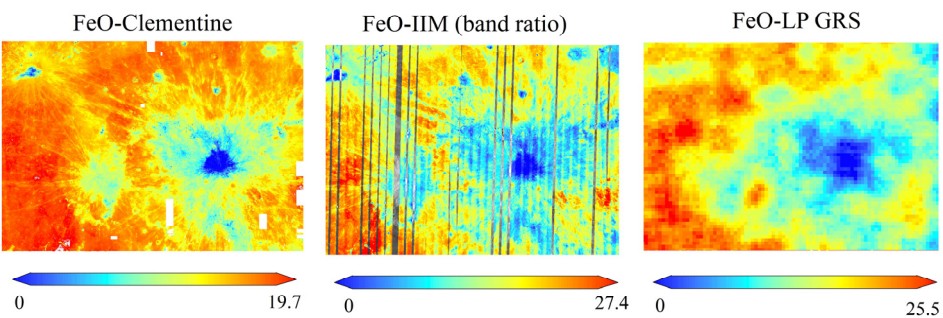

**Figure 15.** Distribution of FeO based on different data/methods.

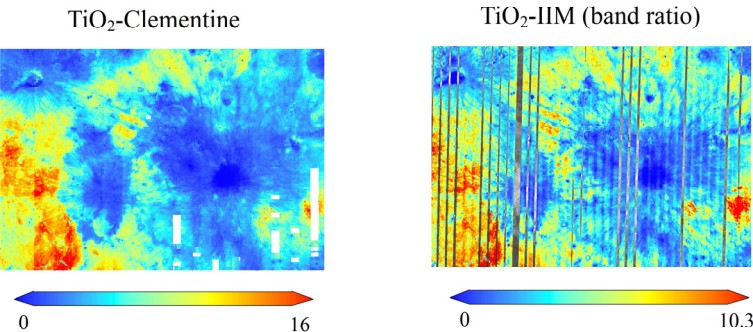

**Figure 16.** Distribution of TiO$_2$ based on different data/methods.

**Table 5.** Correlation coefficients based on different data/methods for FeO inversion.

| Parameters | Statistical Values of Predicted FeO | | | |
|---|---|---|---|---|
| | This Work | Clementine [17] | IIM (Band Ratio) [30] | LP GRS [15] |
| Average | 12.51 | 13.98 | 13.8 | 15.82 |
| Standard Deviation | 4.7 | 5.61 | 6.19 | 6.26 |
| Correlation Coefficients | 1 | 0.91 | 0.89 | 0.9 |

**Table 6.** Correlation coefficients based on different data/methods for TiO$_2$ inversion.

| Parameters | Statistical Values of Predicted TiO$_2$ | | | |
|---|---|---|---|---|
| | This Work | Clementine [17] | IIM (Band Ratio) [31] | LP GRS [15] |
| Average | 2.6 | 4.1 | 2.68 | 2.87 |
| Standard Deviation | 1.36 | 2.94 | 1.64 | 1.67 |
| Correlation Coefficients | 1 | 0.61 | 0.65 | 0.48 |

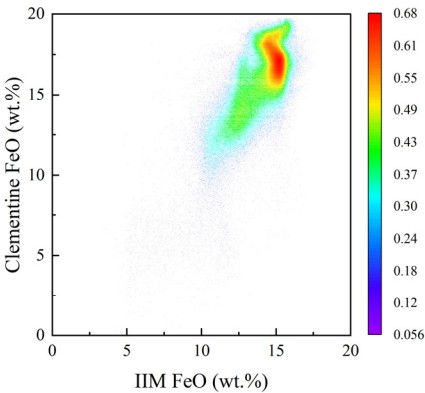

**Figure 17.** Two-dimensional scatter plot of the difference between the Clementine FeO and IIM FeO abundances.

The existence of negative correlations between FeO and Al$_2$O$_3$ and TiO$_2$ and SiO$_2$ was verified. Specifically, two-dimensional density plots of FeO and Al$_2$O$_3$ and two-dimensional density plots of TiO$_2$ and SiO$_2$ were established (Figure 18). The results show that the correlation between FeO and Al$_2$O$_3$ is 75%, and the correlation between TiO$_2$ and SiO$_2$ is 76%. These results are consistent with the previous conclusion that Fe is negatively correlated with Al content and that a decreasing Si phenomenon tends to occur along with the enrichment of Ti elements [64].

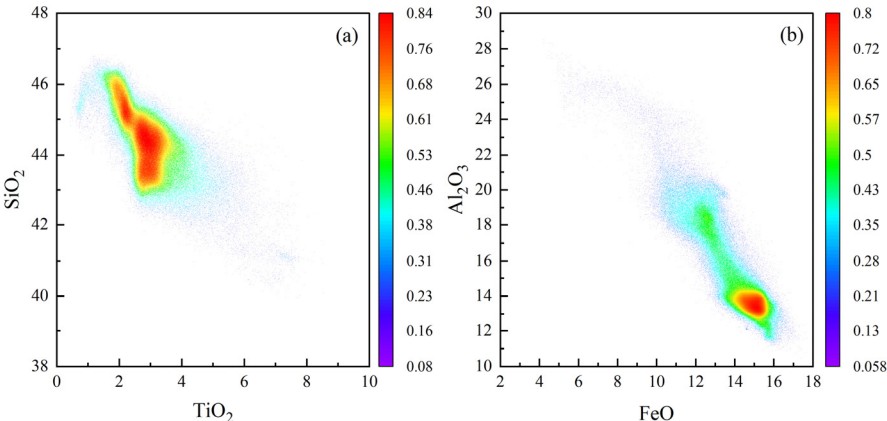

**Figure 18.** (**a**) Two-dimensional density plot between IIM FeO and IIM Al$_2$O$_3$ and (**b**) Two-dimensional density plot between IIM TiO$_2$ and IIM SiO$_2$.

## 4. Discussion

In this paper, a new method of feature band selection combined with Extra-Trees is presented to predict oxide contents on the lunar surface. The results show that the proposed method has good potential for assessing the distribution of oxide contents using spectral features.

### 4.1. Comparison with Other Similar Studies

Previous studies mainly used two bands to retrieve oxide contents according to the absorption characteristics of mineral elements in the visible near-infrared (VNIR) band. For example, 750 nm and 950 nm [17], 757 nm and 891 nm [21], and 757 nm and 918 nm [19] were used to retrieve the FeO content. Additionally, wavelengths of 415 nm and 750 nm [17,67], 531 nm and 757 nm [19], and 321 nm and 415 nm [16] were used to retrieve TiO$_2$ content. Due to the complexity of the lunar soil composition, the ability to describe the nonlinear relationships between spectral features and oxide contents from only two bands is very limited [36], and the accuracy of composition inference is highly dependent on the selected inversion models [68]. In addition, nonchromophore elements do not display absorption characteristics in the VNIR region, which makes it difficult to use most methods in nonchromophore element (Al and Si) inversion. Currently, some scholars [18,20,54] have established the relationships between spectral features and oxide contents based on regression models to invert these oxide contents. Therefore, the inversion of oxide contents on the lunar surface using regression models has become a popular research topic.

Spectral pretreatment methods can improve the correlations between spectral features and oxide contents, as has been well established in previous studies. For example, Li [40] attempted to improve the performance of a model by using a log(1/reflectance) transformation approach. Pieters [26] and Sun et al. [20] applied a band-ratio approach to improve the above correlations. Wu [18] experimentally demonstrated that the correlations obtained considering band differences were superior to those obtained with the band-ratio technique. Therefore, the first-order band-difference transformation method was chosen in this paper to improve the correlations between spectral features and oxide contents. After the preprocessing of spectral data, feature band selection is crucial. However, previous

studies paid little attention to the optimization of feature variables in models. In this study, on the basis of earlier methods [20,37], the Pearson correlation coefficient and SPA are used to select characteristic bands, thus ensuring that high correlations between input spectral features and oxides are retained and reducing the multicollinearity among spectral features. With expanding research, a variety of machine-learning models have been applied to the inversion of lunar surface oxides. The Extra-Trees algorithm selected in this paper can effectively describe the nonlinear characteristics of the studied features and provides strong generalization ability and robustness. This feature band selection approach combined with the Extra-Trees prediction method for predicting oxide contents on the lunar surface yielded reasonable accuracy for the sample test set ($R^2$ values of 0.962, 0.944, 0.964, and 0.86 for FeO, $TiO_2$, $Al_2O_3$, and $SiO_2$, respectively, and RMSEs of 1.028, 0.672, 0.942, and 0.897, respectively) and the regional scale (Apollo17 area; the area near the Copernicus crater).

*4.2. Future Prospects*

(1) The LSCC sample data used in this paper are mainly distributed in the low-latitude area of the lunar nearside, and the number of samples is small; consequently, the distribution characteristics of the oxide contents on the lunar surface cannot be comprehensively reflected, and there are limitations in both quantity and region. This leads to some uncertainty in the prediction results. In the future, more samples should be obtained to supplement the sample data limitations in certain regions of the Moon, such as at middle and high latitudes, and increase the number of samples, which is expected to improve the accuracy of chemical abundance distribution characteristics.

(2) There are spectral anomalies at the edges of different orbit images of IIM hyperspectral data, and factors such as solar azimuth and topographic relief will lead to shadows in the images, which will inevitably increase errors in the inversion results. In future research, we should consider how to mitigate the spectral anomalies and topographic shadows in IIM data or explore the potential of using higher-quality and higher-resolution remote sensing data for inversion, such as the $M^3$ data obtained with the Indian satellite Chandrayaan-1 and the MI data obtained with Kaguya in Japan, to improve the prediction accuracy of the model.

(3) The continuity of hyperspectral data greatly enriches the amount of information available in remote sensing data, but it can also lead to issues such as information redundancy and high correlations between bands. Therefore, determining how to obtain the best combination of sensitive bands is an important step in the application of hyperspectral data. The sensitive band screening method applied in this paper provides a reference for other chemistry inversion research. In the future, more band screening algorithms can be applied for feature selection with lunar hyperspectral data to reasonably select the best number and combination of bands and improve accuracy.

(4) In the era of big data, big data theory and technology are important tools for solving practical problems. The application of machine learning and deep learning algorithms for lunar chemistry inversion is still in its infancy. Although the Extra-Trees model developed in this paper provides good prediction ability, there is still room for further improvement. Thus, a future development direction is to complete lunar surface oxide inversion through better machine learning and deep learning methods.

## 5. Conclusions

The plan to return to the Moon in the new era is to develop and utilize lunar resources, establish a lunar base, and use the Moon as a springboard to gradually carry out deep space exploration. Through comprehensive remote sensing exploration of the Moon to analyze and assess the distribution of lunar mineral resources, it can provide a long-term stable resource reserve for the sustainable development of human society. In this paper, using IIM data, the original reflectance as the initial input and the oxide contents of LSCC samples as the desired outputs, we use a combination of the mathematical transformation of spectral data (first-order difference method), a feature band selection technique (with

the SPA) and a machine learning modeling method (Extra-Trees). The correlations between reflectance and oxide contents are improved to identify the difference bands most sensitive to oxide levels and construct a prediction model with excellent inversion ability for lunar surface oxide (FeO, $TiO_2$, $Al_2O_3$, and $SiO_2$) contents. This approach can solve the problems of low accuracy and poor generalizability encountered with traditional modeling methods based on raw spectral data. The main conclusions of this paper are as follows.

(1) The correlation calculations for the original bands and all band differences in the IIM spectral range separately show that considering the band differences can enhance the correlations between reflectance and oxide contents. Thus, the best combination of spectral bands can be used in subsequent modeling, and the accuracy of the model can be improved. Moreover, the calculated interelement correlations suggest that Fe is negatively correlated with Al and that Si depletion is often accompanied by the enrichment of Ti.

(2) In total, 325 band differences were initially screened using Pearson correlation coefficients, and then secondary downscaling screening was performed according to BKM combined with SPA. Consequently, 17, 5, 8, and 30 feature bands were retained for the four oxides (FeO, $TiO_2$, $Al_2O_3$, and $SiO_2$, respectively) for modeling. The SPA encompasses most of the spectral information associated with samples, effectively reduces the complexity of modeling and reduces the covariance interference among spectral features.

(3) Machine learning algorithms are increasingly integrated with lunar surface chemistry inversion in the big data era. Big data undoubtedly provide new data-driven research methods and can effectively overcome the relatively limited numbers of lunar samples. Moreover, the high dimensional and complex nonlinear relationships between oxide contents and spectral reflectance can be better described. In this paper, we apply the Extra-Trees algorithm for chemistry inversion to predict the distribution of chemistry on the lunar surface. The results show that the $R^2$ values of the test sets for FeO, $TiO_2$, $Al_2O_3$, and $SiO_2$ are 0.962, 0.944, 0.964, and 0.860, respectively, and the RMSE values are 1.028, 0.672, 0.942, and 0.897, respectively, improving the modeling accuracy over the original bands.

(4) The average contents of the four oxides (FeO, $TiO_2$, $Al_2O_3$, and $SiO_2$) in the region near the Copernicus crater are 12.51 wt.%, 2.60 wt.%, 13.56 wt.%, and 39.53 wt.%, respectively. The oxide content distributions display obvious variations, and the frequency histograms show a clear bimodal distribution. The comparison of the present result with representative models shows that the model in this paper provides good agreement in the inversion of oxide abundance, thus providing a new idea and method for the inversion of oxide abundance.

**Author Contributions:** Conceptualization, S.W. and J.C.; methodology, S.W.; software, S.W.; validation, S.W., J.C., and L.L.; formal analysis, S.W.; resources, C.Z. and R.H.; data curation, S.W., J.C., and C.Z.; writing—original draft preparation, S.W.; writing—review and editing, S.W., J.C., and L.L.; visualization, S.W., C.Z., and Q.Z.; supervision, S.W., L.L., and R.H.; funding acquisition, J.C. All authors have read and agreed to the published version of the manuscript.

**Funding:** This research was funded by the Geological Survey Project of China Geological Survey, grant No. 42932022001, and the Natural Science Basis Research Plan in Shaanxi province of China, grant No. 2020JQ-742.

**Data Availability Statement:** Chang'E-1 IIM data are provided by "The Science and Application Center for Moon and Deep Space Exploration" at http://Moon.bao.ac.cn/.

**Acknowledgments:** The authors gratefully acknowledge Zongcheng Ling for providing the improved Chang'E-1 IIM imagery and Chang Liu and Yiwen Pan for their guidance in mineralogy. The authors also thank the cartographers who compiled the geological map of the Moon.

**Conflicts of Interest:** The authors declare no conflictd of interest.

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
