# Peer review of "Quantitative Inversion of Lunar Surface Chemistry Based on Hyperspectral Feature Bands and Extremely Randomized Trees Algorithm"

_remotesensing, doi:10.3390/rs14205248_

Round 1

Reviewer 1 Report

This is a great manuscript very clear and well-written and well-structured! The application of Pearson correlation analysis plus Extreme radomized trees algorithm to the determination of lunar surface oxides by inversion from hyperspectral data is a very interesting and powerful technique which will be of interest to inversion problems from hyperspectral data far more generally. The description of the method, of the data and of the results is scientifically sound and convincing. The conclusions are insightful and adequate. Citations are appropriate. The manuscript should be published in its current form.

Author Response

Thank you so much for your very careful review of our manuscript and positive evaluation to our work.

Reviewer 2 Report

Derive elemental distribution of the Moon by remote sensing data set is meaningful for geologic studies of the Moon. My comment can be found in the attachment as follows.

Author Response

We sincerely appreciate the time and thoughtful comments devoted by the reviewer. We have carefully revised our manuscript according to the reviewers’ comments and suggestions. Please see the attachment.

Reviewer 3 Report

Overview: I recommend rejecting this paper. That said, the authors could re-write this paper, include a new co-author with lunar mineralogy expertise, and consider submitting that revised paper. The current paper makes some naïve and sometimes incorrect statements about lunar mineralogy while at the same time going into extreme depth with big data analytics methods without explaining some of the fundamentals. As such it is severely unbalanced. The paper reads as if the authors are experts in data analytics and hoping to apply that method to problems they know very little about. They would thus be wise to collaborate with experts in the Moon’s geology and mineralogy on any submission of work based on this initial manuscript. Finally, the authors would be wise to be more explicit in what science question they are trying to address: how will a refined knowledge of compositional variation discriminate amongst competing hypotheses for lunar crustal formation, for instance? Or about impact regolith gardening? Or, is this paper about a new data analytics technique and the lunar geology application is incidental? In any case, and whatever the specific science question is, the authors should think carefully in their writing of a new paper. Regarding true lunar oxides, most notably ilmenite (FeTiO3), the authors should mention its utility in In-Situ Resource Utilization (ISRU) in extracting oxygen and metal for human lunar infrastructure as a primary motivator. In fact, I would recommend this paper be restructured with finding small (km-scale) ilmenite deposits as the primary motivator.

Abstract: The authors begin with a major mis-characterization of lunar elemental abundance: While FeO and TiO2 (as well as FeTiO3, which is not mentioned by the authors even in the abstract [though its English name ilmenite is mentioned in the text]), Al2O3 and SiO2 are not in the oxide form but rather bound as silicates. Let me state that again: Al2O3 and SiO2 are not oxides—they are silicates as found on the Moon in that they are (nearly) always found bound in more complex minerals, often anorthite CaAl2Si2O8, pyroxene (Fe,Mg)2Si2O6, and olivine (Fe,Mg)2SiO4. In fact, ilmenite FeTiO3 is the most common true oxide on the Moon; Some of the mentioned FeO and TiO2 are incorporated in to the olivine and pyroxene silicate mineralogies, as I show above. I would direct the authors to the Lunar Sourcebook, maintained by the NASA-funded Lunar and Planetary Institute, as an excellent primer on lunar mineralogy: https://www.lpi.usra.edu/publications/books/lunar_sourcebook/ . Also, Chapter 2 of the recent Planetary Science Decadal Survey contains an excellent overview of mineralogy on the Moon: https://nap.nationalacademies.org/read/26522/chapter/4.

Line 44: “SiO2 plays an important role in studies of the classification and genesis of lunar rock.” This statement is so naïve as to undermine the trustworthiness of this paper. SiO2 is in nearly every rock and mineral grain on the Moon as the fundamental component of silicate minerals. Inclusions of statements like this gives me pause and makes me doubt that the authors really understand mineralogy—lunar or terrestrial.

Lines 52-43: If this is as paper on a new technique for analyzing lunar spectral data, then it is not necessary to give the quantum mechanical basis for spectroscopy. Again, such statements undermine the credibility of the authors.

Line 107: Please include some example citations of recent uses of big data.

Line 119: We don’t have samples from near Copernicus Crater to test against the authors’ data inversion model, so their claim is unsubstantiated that “the model performs well in the inversion of lunar surface oxide abundance.” Granted, their model may stand up against laboratory mixtures.

Line 163: The authors should check to see if the URL to the Brown University relab site is still working; the web page would not load for me.

Line 443: I have never heard of a Giant Mare Tectonic Unit and the authors do not provide a citation. Did they invent this term? I’m not sure it’s helpful—what are they trying to imply by using the word, “Tectonic?”

Figure 13: Did the authors get permission to use this figure from another publication? Also, the legend is illegible. Why are they even showing this figure? I suggest removing it; or if they keep it, state that it’s reproduced from a previous publication and to state that they got permission (if indeed they did so) and to make the legend larger.

Line 473: Is improving from previous publications the point of this? Is a 9% improvement over Clementine really worth all this trouble? Also, previous work from Clementine, Moon Mineralogy Mapper, and the color Lunar Reconnaissance Orbiter Wide Angle Camera should have been covered in the introduction. As it stands I don’t know the reason for this work.

Author Response

(The authors gave the same response as above.)
